# Evidence for Radiation Therapy in Stage III Locoregionally Advanced Cutaneous Melanoma in the Post-Immunotherapy Era: A Literature Review

**DOI:** 10.3390/cancers16173027

**Published:** 2024-08-30

**Authors:** Jennifer Zhou, Evan Wuthrick

**Affiliations:** 1Morsani College of Medicine, University of South Florida, Tampa, FL 33602, USA; 2Department of Radiation, H. Lee Moffitt Cancer Center and Research Institute, Tampa, FL 33612, USA; evan.wuthrick@moffitt.org

**Keywords:** melanoma, radiation therapy, immune checkpoint inhibitors, stage III melanoma

## Abstract

**Simple Summary:**

Melanoma is the deadliest of all skin cancers. Therapy paradigms in cutaneous melanoma have changed significantly over the past 20 years, following both changes in the initial staging work-up of melanoma as well as in the available therapies. Regional recurrences after initial surgery are normally treated with nodal basin resection. However, some regional recurrences may present as bulky masses that can pose significant challenges in the treatment and control of the disease. It is well established that radiation therapy (RT) plays a role in locoregional control of malignancies in general. This review will attempt to discuss evidence for radiation therapy’s role in reducing regional recurrence in the adjuvant setting in melanoma, identify adjuvant systemic therapy options, discuss systemic therapy–radiation therapy combinations, and identify paradigms and emerging evidence for the use of radiation therapy after neoadjuvant systemic therapy strategies are employed.

**Abstract:**

In the landscape of Stage III locoregionally advanced cutaneous melanoma treatment, the post-immunotherapy era has sparked a number of questions on the management of the nodal basin. However, much of the available literature is not focused on radiation therapy as an adjuvant therapy. This literature review aims to illuminate the evidence surrounding radiation therapy’s potential to mitigate regional recurrences in the adjuvant setting for melanoma. Additionally, it seeks to identify adjunct systemic therapy options and explore the synergy between systemic therapy and radiation. Despite strides in surgical techniques and systemic therapies, controlling regional Stage III melanoma remains a formidable clinical hurdle. While historical data strongly suggest the efficacy of adjuvant radiation therapy in reducing regional recurrence risk, its evaluation predates the advent of MAPK pathway inhibitors and robust immunotherapy options. Notably, clinical trials have yet to definitively demonstrate a survival advantage with adjuvant radiation therapy. Additional research should focus on refining the definition of high risk for regional recurrence through gene expression profiling or tumor immune profiling scores and elucidate the optimal role of adjuvant radiation therapy in patients treated with neoadjuvant systemic therapy.

## 1. Introduction

The incidence of cutaneous melanoma continues to steadily rise and accounts for the majority of skin-cancer-related deaths [1]. Approximately 80% of melanoma patients present with clinically localized disease that can usually be managed with wide local excision and draining nodal basin evaluation. Among clinically node-negative patients, nodal evaluation consists of sentinel lymph node biopsy if the primary Breslow depth is >0.7 mm. Clinically node-positive patients are traditionally managed with complete lymph node dissection (LND) [1]. However, up to 12.6% of melanoma patients will experience a locoregional recurrence of their melanoma in the nodal basin after nodal resection within 10 years [2,3]. Resectable, isolated regional recurrences are managed by nodal basin dissection [4]. However, some of these recurrences appear as multiple bulky masses. These locoregional recurrences are sometimes unresectable, symptomatic masses that are challenging to treat and control [4]. Additionally, advanced locoregional melanoma poses a significant detriment to patients’ quality of life and emotional well-being [5].

There are many factors that influence the risk of regional melanoma recurrence in the nodal basin following lymphadenectomy. Multiple studies have demonstrated associations between locoregional recurrence of melanoma and extracapsular extension, size of largest involved lymph node, multiple lymph node involvement, site of regional lymph node metastasis, non-sentinel lymph node metastases [6], and number of lymph nodes dissected [7,8,9,10,11,12]. In a study looking at nodal basin recurrence following lymph node dissection, nodal basin recurrence at 10 years was noted to be 43%, 28%, and 23% in the cervical, axillary, and inguinal regions, respectively (*p* = 0.08) [9]. A recent study by Uppal et al. also reported an increased risk of regional melanoma recurrence after LND for clinically palpable disease compared to LND for pathologically identified disease [13].

Additionally, the prognosis in patients with a positive sentinel lymph node biopsy varies based on the pathologic features of the sentinel lymph node. Currently, the strongest prognostic and predictive parameter for survival is the Rotterdam–Dewar combined (RDC) criteria, which take into account both metastasis burden and position of metastasis in sentinel lymph nodes [14]. On multivariate analysis, patients noted to have RDC criteria > 1.0 mm and both subcapsular and non-subcapsular lymph node involvement were noted to have significantly increased melanoma-specific mortality compared to those with RDC criteria < 0.1 mm and only subcapsular extension (HR, 6.17; 95% CI, 1.95–19.45; *p* < 0.001) [14]. The RDC criteria have the limitation of not predicting the anatomic location of future recurrences (locoregional versus distant). However, these data do indicate that node-positive patients with certain features may have an increased risk of recurrence and a worse prognosis. Several recent landmark surgical trials have informed the management of non-metastatic melanoma. The Multicenter Selective Lymphadenectomy Trial (MSLT) I randomized patients into wide local excision with or without sentinel lymph node biopsy in patients with clinically negative nodal basin. MSLT I demonstrated that sentinel node biopsy-based management significantly increased 10-year disease-free survival, with 71.3% of patients remaining disease free at 10 years in the intermediate thickness group compared to 64.7% in the control arm [15]. Superiority of the sentinel lymph node biopsy was also seen in the thick melanoma group, with 50.7% of patients in the experimental arm remaining disease free at 10 years compared to 40.5% in the control arm [15].

MSLT II established that complete lymph node dissection (CLND) following positive sentinel lymph node biopsy had no significant effect on overall survival in patients with melanoma [6]. Based on this study, Verver and colleagues developed a nomogram (see Figure 1) for predicting clinical outcomes of patients with positive sentinel lymph node melanoma without the need for complete lymph node dissection with a C-index in an external validation cohort for recurrence, distant metastasis, and overall mortality of 0.70, 0.72, and 0.74, respectively [16].

Though MSLT II did not demonstrate survival outcome differences with complete lymphadenectomy, higher rates of regional recurrence were seen in the sentinel-lymph-node-only group [6,17]. An additional publication looking at the MSLT II population identified risk factors associated with regional recurrence, including the following: age (older than 50 years old), ulceration, Breslow thickness greater than 3.5 mm, nonaxillary basin, and a tumor burden of maximum diameter greater or equal to 1 mm and/or a greater than 5% metastasis area [17]. Among those with four or more risk factors, the risk of regional recurrence was greater than 40%, which is high enough that additional nodal control with either surgery or radiation therapy may improve outcomes. This finding is supported by both DeCOG-SLT trials and follow-up analyses of MSLT II, which found a decrease in nodal recurrence in patients who underwent CLND compared to patients who underwent active observation of 57.1% and 67%, respectively [18,19].

After the publication of these data by MSLT II, National Comprehensive Cancer network® (NCCN®) published new guidelines regarding treatment of melanoma, suggesting observation over CLND after sentinel lymph node biopsy [3]. As more patients opt for observation over CLND, we can extrapolate, based on the results of MSLT II, that there will be an increasing population of patients with nodal recurrence of melanoma after surgical management.

It is well established that radiation therapy (RT) plays a large role in locoregional control of malignancies in general. This review aims to demonstrate evidence for radiation therapy’s role in reducing regional recurrence in the adjuvant setting in melanoma, discuss systemic therapy–radiation therapy combinations, and discuss paradigms and emerging evidence for the use of radiation therapy after neoadjuvant systemic therapy strategies are employed.

## 2. Evidence for Radiation Therapy’s Role in Regional Recurrence Risk Reduction

Radiation therapy’s ability to control locoregional spread of malignancies in general is well known. As such, multiple retrospective studies have been published on the topic of radiation therapy for the management of lymph node basin recurrences from melanoma [7,10,11,12,20].

The largest retrospective study that suggested a role is Agrawal and colleagues’ study of 615 melanoma patients from two institutions in 2009 [20]. All patients in this study had high risk nodal disease in either axillary, cervical, or inguinal basins. Of these, 509 patients underwent radiation therapy in their nodal basin following lymphadenectomy, while the other 106 patients simply underwent observation following lymphadenectomy. The patients in the radiation therapy arm received 30 Gy in five fractions twice weekly (6 Gy/fraction). The analysis of the outcomes showed a significant improvement in locoregional control and disease-specific survival (DSS) in the radiation therapy arm compared to the observation arm at 5-year follow-up. The regional control rate at 5 years was 89.8% in the radiation therapy arm versus 59.4% in the observation arm (*p* < 0.001). Additionally, disease-specific survival was significantly improved from 30% in the observation arm versus 51% in the radiation therapy arm (*p* < 0.0001).

However, patients in the radiation therapy arm were noted to suffer from treatment-emergent adverse effects—the most important of which is lymphedema—at a significantly higher rate of 20% versus 13% in the control arm (*p* < 0.004). Studies by Ballo et al. also supported radiation therapy’s role in regional control of cervical (94% at 10 years with adjuvant RT) [10], axillary (87% at 5 years) [12], and inguinal lymph nodes (74% at 3 years) [11] with adjuvant RT. Based on these findings, Ballo et al. also published their indications for radiation therapy, including ECE, lymph nodes measuring 3 cm in size or larger, the involvement of multiple lymph nodes, recurrent disease, or any patient having undergone a therapeutic LND [10,11,12].

There is currently only one phase III randomized trial looking at the effects of radiation therapy versus observation following lymphadenectomy in patients with high-risk nodal disease—the TROG 02.01 study [21,22]. TROG 02.01 was a phase III prospective study that included 217 patients with high-risk nodal melanoma who were randomized to either an adjuvant radiation (experimental) or observation (control) arm. A total of 109 patients were treated with radiation therapy, and 108 patients were observed for recurrence. For patients in the experimental arm, a dose of 48 Gy was given in 20 fractions over 4 weeks (2.4 Gy/fraction) and within 12 weeks of lymph node dissection. Researchers in this study found a significant difference between both groups in terms of regional control of lymph node basin, with a 3-year recurrence rate of 19% in the radiation therapy arm versus 31% in the observation arm (20 vs. 34 relapses; HR, 0.56; 95% CI, 0.32–0.98; *p* = 0.041) (Figure 2).

Despite this, overall survival was not noted to be significantly improved in the experimental arm compared to the control arm (59 vs. 47 deaths; HR, 1.37; 95% CI, 0.94–2.01; *p* = 0.12) (Figure 3). Additionally, researchers continued to follow this population of patients and published long-term outcomes in 2015 [22]. Similar to the original TROG 02.01 study, they found a significant increase in regional recurrences in the observation arm compared to the experimental arm (HR, 0.52; 95% CI, 0.31–0.88; *p* = 0.023) but no significance in overall survival between either group (HR, 1.27; 95% CI, 0.89–1.79; *p* = 0.21). Contrary to Agrawal and colleagues, the researchers discovered no significant difference in relapse-free survival between the experimental and control arms (HR, 0.89; 95% CI, 0.65–1.22; *p* = 0.51), although their study was relatively small and not powered to assess endpoints such as relapse free survival or overall survival. Similar to the prospective randomized data, several retrospective studies note that radiation therapy may increase regional control of melanoma but might not influence overall survival.

There have been a few theories proposed to explain this phenomenon. The most parsimonious theory is that RT has no effect on distant metastases and therefore no effect on melanoma-specific survival. If this is the case, then post-operative RT or any type of adjuvant RT should be used only to reduce the risk of regional recurrences and to prevent potential toxicities associated with salvage therapies. Toxicity reports of adjuvant therapy to prevent nodal recurrences should be considered as of utmost importance in this case.

Another possible reason that radiation therapy may improve regional control but not influence overall survival is that regional RT improves overall survival in a small and, as of yet, unidentified population of patients. This is supported by Strom et al., who retrospectively looked at a cohort of 410 patients from one institution with AJCC Stage III melanoma without evidence of metastases beyond the lymph nodes, who either did or did not receive RT after lymphadenectomy, to clarify the role of regional RT with respect to individual risk factors [7]. They confirmed in their study that patients were at a significantly increased risk of nodal recurrence at 5 years if they had node-positive melanoma and either palpable lymph nodes (HR, 2.40; 95% CI, 1.32–4.36; *p* = 0.004) or ECE (HR, 2.17; 95% CI, 1.17–4.01; *p* = 0.01). Radiation therapy was noted to have high regional control rates at 1-, 2-, and 5-years post-treatment (96.3%, 95.0%, and 95.0%, respectively) compared with treatment without regional RT (91.7%, 87.4%, and 83.3%, respectively; *p* = 0.036). Among the 213 patients with the highest risk of regional recurrence, researchers noted that RT following lymphadenectomy was associated with a significantly lower risk of additional treatment for regional recurrence at 1-, 2-, and 5-years post-treatment (2.7%, 2.7%, and 2.7% vs. 12.7%, 18.1%, and 26.0%, respectively; HR, 0.12; 95% CI, 0.03–0.50; *p* = 0.004).

One potential method of identifying the patients at highest risk of regional recurrence could be through gene expression profiling of tumor tissue. In Strom et al., they also looked at patients’ radiosensitivity index (RSI) gene expression signature (GES) and found that patients who had a low RSI GES (radiosensitive) that received RT had significantly increased survival than those with a high RSI GES (less radiosensitive), with 1-, 2-, and 5-year estimated survival rates of 100%, 100%, and 75% compared with 85.7%, 14.3%, and 0%, respectively (HR, 10.68l 95% CI, 1.24–92.14). Despite this and consistent with previous studies, no significant difference was seen in survival by RSI GES status among low RSI index patients (n = 12) without adjuvant RT. Estimated 1-, 2-, and 5-year survival rates were 100%, 91.7%, and 56.3% among low RSI patients, compared with 84.2%, 63.2%, and 27.1% for high RSI patients (n = 19; *p* = 0.19). Perfecting nomograms or other models and adding gene expression data may refine high-risk groups used in the future.

Because overall survival as well as RFS have not be proven to be affected by adjuvant radiation therapy, the use of post-operative radiation therapy for Stage III melanoma remains controversial. In the current NCCN guidelines on melanoma, adjuvant radiation therapy for Stage III melanoma is a category 2B recommendation [23]. This category indicates that this recommendation lacks uniform consensus, and it is based on lower-level evidence.

Moreover, the guidelines state that adjuvant RT after LND should be considered only if specific high-risk criteria are met. These high-risk criteria vary depending on the nodal station involved, that is the parotid, cervical, axillary, or inguinal stations. The criteria include extranodal tumor extension (ECE) and LDH >1.5× upper limit of normal AND/OR parotid: ≥1 involved node, any size of involvement; cervical: ≥2 involved nodes and/or ≥3 cm tumor within a node; axillary: ≥2 involved nodes and/or ≥4 cm tumor within a node; inguinal: ≥3 involved nodes and/or ≥4 cm tumor within a node. There is currently insufficient published data to give recommendations on adjuvant radiation therapy to epitrochlear or popliteal lymph node basins after dissection. In clinical practice in the United States, it remains widely heterogenous which Stage III melanoma patients are offered adjuvant radiation therapy.

## 3. Current Adjuvant Systemic Therapies

There are two molecular categories of Stage III melanoma: those with MAPK pathway mutations and those without. The main driver mutations in the MAPK pathway in melanoma are the BRAF and NRAS genes. Roughly around 40–50% of cutaneous melanoma studied have a BRAF mutation [24,25]. Systemic therapies discussed further in this review target either the MAPK pathway or previously established immune checkpoint inhibition pathways such as the CTLA-4 (cytotoxic T-lymphocyte associated protein 4) and PD-1 (programmed cell death protein 1) receptor pathways.

Ipilimumab is a monoclonal antibody immune checkpoint inhibitor that blocks CTLA-4, a receptor on T cells that inhibits activation of T cells and subsequent death of malignant cells. Ipilimumab has demonstrated continued benefit as an adjuvant therapy for resected Stage III melanoma [26]. EORTC 18071 was a phase III, randomized, double-blinded study looking at 951 patients with fully resected Stage III melanoma who were treated with either 10 mg/kg ipilimumab or a matched placebo every 3 weeks for four doses, followed by one dose every 3 months up to 3 years [27]. In the long-term follow-up analysis paper published in 2019, researchers found that OS was significantly longer for ipilimumab versus placebo (HR, stratified by stage, 0.73; 95% CI: 0.60–0.89; *p* = 0.002). Additionally, locoregional recurrences as the only type of recurrence occurred in 17.6% of patients treated with ipilimumab, compared to 21.6% in patients treated with placebo.

PD-1 is an immune checkpoint receptor present on activated T-cells. Anti-PD-1 therapy attempts to block the upregulated ligands, PD-1 and PD-2, on cancer cells from binding to the receptor on T-cells, thus leading to T-cell activation and the appropriate anti-tumor response. Anti-PD-1 therapeutic agents pembrolizumab and nivolumab are the only two agents with high-level data in the adjuvant setting for melanoma. Prompted by the significant improvement in OS, PFS, and overall response rate of pembrolizumab in the KEYNOTE-006 trial [28], KEYNOTE-054 was a prospective phase III trial looking at pembrolizumab as an adjuvant therapy in Stage III melanoma [29]. A total of 1019 patients were given 200 mg pembrolizumab every 3 weeks for a year vs. placebo after complete surgical excision of malignant tissue. Interestingly, pembrolizumab was noted to have a non-significant decrease of 27% (HR, 0.73; 95% CI, 0.54–1.00) in locoregional recurrences as the first RFS event at 3.5 years, indicating that patients were more likely to experience decreased metastasis-free recurrence as opposed to decreased locoregional recurrences. Grade 3+ adverse events were reported in 14.7% of patients in the pembrolizumab group compared to 3.4% in the placebo arm, which remained unchanged after the initial study was published in 2018 [30].

Another landmark prospective phase 3 trial involving anti-PD-1 therapy was Checkmate-238, which looked at 906 patients with Stage IIIB-IV completely resected melanoma who were treated with either nivolumab or ipilimumab [31]. Patients were given either 3 mg nivolumab every 2 weeks or ipilimumab 10 mg every 3 weeks × 4 doses and then every 4 weeks. Nivolumab was associated with a statistically significantly longer RFS at 4 years, with 53% of patients in the nivolumab arm remaining recurrence free compared to 44% in the ipilimumab arm (HR, 0.71; 95% CI, 0.60-0.86; *p* = 0.0003). Grade 3 adverse events were noted in 14.4% in the nivolumab group versus 45.9% in the ipilimumab group. Late-emergent adverse events of grade 3+ were reported in an additional 1% of patients in the nivolumab group compared to 2% in the ipilimumab arm.

Dabrafenib is a BRAF V600 inhibitor approved for metastatic melanoma in 2013, after it was shown to significantly improve PFS over dacarbazine in patients with a BRAF V600 mutation and untreated metastatic disease [32]. Trametinib is a MEK1/MEK2 inhibitor that was approved by the FDA in 2013 initially for patients with metastatic disease and a BRAF V600 mutation [33]. Dual agent MAPK-pathway inhibition is shown to be more efficacious than single agent MAPK-pathway inhibition in metastatic melanoma [34]. A combination of dabrafenib and trametinib was approved for adjuvant usage by the FDA for patients with Stage III melanoma in 2018. A combination of 150 mg dabrafenib twice daily and 2 mg trametinib once daily was shown to reduce distant metastases in 870 randomized patients with Stage III resected melanoma by 45% at 5 years (HR, 0.55; 95% CI, 0.44–0.70) [35]. Long-term follow-up of patients treated with dabrafenib and trametinib versus placebo indicated that patients experienced a decrease in locoregional recurrences; a time-to-event statistical comparison was not reported [36]. Grade 3+ adverse events were noted in 36% of patients in the experimental arm vs. 10% in the placebo arm.

BRIM 8 was an international, double-blinded, placebo-controlled phase III trial looking at another adjuvant BRAF V600 inhibitor—vemurafenib—in 498 patients with AJCC Stage IIC, IIIA, IIIB, or IIIC fully resected melanoma [37]. A total of 250 patients in two cohorts—73 patients in Stage IIIC (Cohort 2) and 157 patients in Stage IIC-IIIB (Cohort 1)—received 960 mg vemurafenib twice daily for 52 weeks. The rest of the patients in the study received a matching placebo. The primary endpoint of disease-free survival was not reached in cohort 2. Because of the prespecified hierarchical testing of cohort 2 before cohort 1, and because the primary endpoint was not met in cohort 2, the analyses in cohort 1 cannot be regarded as significant. However, it was noted that a numerical benefit in disease free survival was associated with vemurafenib compared to placebo in patients with Stage IIC-IIIB melanoma. Patients in the vemurafenib arm reported increased grade 3+ adverse events compared to the placebo arm (57% vs. 15%, respectively). A more recent analysis of the patient population treated in BRIM 8 indicated that vemurafenib was more effective in reducing recurrence in patients with <5% PD-L1+ immune cells (IC) (HR, 0.36; 95% CI, 0.24-0.56) and <1% CD8+ T-cell count (HR, 0.56; 95% CI, 0.34–0.92) compared to patients with ≥5% PD-L1+ IC (HR, 0.99; 95% CI, 0.58–1.69) and ≥1% CD8+ T-cell count (HR, 0.77; 95% CI, 0.48–1.22) [38].

## 4. Current Systemic Therapy Strategies and Radiation Therapy as an Adjuvant Therapy

Radiation therapy has been associated with increased anti-tumor response in areas outside of the radiation field—an effect termed the “abscopal effect”. Some preclinical and phase I-II trials have shown benefits of radiation therapy in addition to systemic immune checkpoint inhibitors in other malignancies [39,40,41]. More specifically, radiation therapy has been shown to activate T-cells in multiple preclinical trials via a variety of ways, making them more primed to target and destroy cancer cells [40]. Combined with immune checkpoint inhibitors, these activated T-cells are hypothesized to be more sensitive to cancer cells everywhere. Despite this, there are no prospective studies looking at current systemic therapy strategies + RT as an adjuvant therapy for melanoma. Multiple phase I or phase II studies are currently being conducted to further investigate radiation therapy’s role with systemic therapies in patients with advanced melanoma (see Table 1). These studies are novel in that they are evaluating the effects of radiation therapy in combination with immunotherapy in advanced melanoma and at a stage where radiation therapy is traditionally reserved for local palliation of symptoms. 

One recent study has retrospectively looked at 109 patients with Stage III melanoma who received systemic therapies as well as adjuvant radiation therapy after lymph node dissection [42]. 109 patients from four institutions who all received systemic therapy either alone or in combination with radiation therapy were compared for lymph node basin (LNB) recurrences. Researchers noted that in the 3-year cumulative incidence of LNB recurrences between these cohorts, none of the patients who received systemic therapy (ST) with concurrent radiation therapy noted recurrences, while the incidence of recurrence for the ST alone group was 27.0% (n  =  8; 95% CI 12.7–33.4%), but this difference was not statistically significant (*p*  =  0.21). Researchers noted that this was most likely due to the low number of patients who did have LNB recurrences.

## 5. Use of Adjuvant Radiation Therapy after Poor Response to Neoadjuvant Immunotherapy or MAPK Pathway Inhibitors

More recently, the idea of using immunotherapy or MAPK pathway inhibitors prior to surgical treatment of clinically node-positive melanoma—neoadjuvant therapy—has become increasingly common. It is thought to increase patient survival through a few advantages: an increased tumor antigen load at the time of treatment, which is thought to better educate the antitumor specific immune response; an increased likelihood of reducing the tumor burden to improve surgical morbidity; the ability to address potential microscopic dissemination of malignancy earlier in the treatment course; and, lastly, the capacity to measure the pathologic response as a biomarker of response to the therapy applied.

There are a few clinical trials looking at the efficacy of neoadjuvant immunotherapies/MAPK pathway inhibitors that are currently ongoing. Initial results appear promising. A phase 1b trial (NCT02434354) looked at a single 300 mg dose of neoadjuvant pembrolizumab prior to resection 3 weeks after in 27 patients with resectable Stage III or Stage IV melanoma [43]. A total of 29.6% (n = 8) of patients noted a major pathological response (defined as ≤10% of viable tumor cells) after one dose of pembrolizumab.

Additionally, a prospective phase II study (NCT02519322) included 23 patients with resectable Stage III or oligometastatic Stage IV melanoma to receive neoadjuvant 3 mg/kg nivolumab vs. combination 3 mg/kg ipilimumab and 1 mg/kg nivolumab prior to surgery [44]. After surgery, patients were given 1 year of adjuvant therapy. A total of 12 patients were given nivolumab monotherapy, and 11 patients were given a combination of both ipilimumab and nivolumab. A 25% RECIST overall response rate and pathologic complete response (pCR) was observed in patients given neoadjuvant nivolumab, compared to a 73% RECIST overall response rate and a 45% pCR in patients given combination nivolumab and ipilimumab. Of note, grade 3+ adverse events were noted in 73% (n = 8) of patients given combination neoadjuvant therapy, compared to 8% (n = 1) in patients given monotherapy nivolumab. An additional study (OpACIN-neo) was conducted to find the optimal dosing of nivolumab and ipilimumab in the neoadjuvant setting (NCT02977052) [45]. Patients with Stage III melanoma were randomized into three arms: ipilimumab 3 mg/kg and nivolumab 1 mg/kg, ipilimumab 1 mg/kg and nivolumab 3 mg/kg, and sequential ipilimumab 3 mg/kg followed by nivolumab 3 mg/kg. Based on response and toxicities noted, nivolumab 3 mg/kg and ipilimumab 1 mg/kg was determined to be the best regimen to study in future trials.

Based on the safety data obtained from OpACIN-neo, PRADO was an open-label phase II trial that examined 99 patients with Stage IIIB or IIIC melanoma who also had a measurable index lymph node (ILN) [46]. Patients were given 3 mg/kg nivolumab and 1 mg/kg ipilimumab every 3 weeks for up to two doses followed by resection of the marked ILN. In the entire study population, pCR or near pCR were noted in 61% of patients (n = 60). Despite this, pathologic non-response (pNR)—defined as >50% viable tumor—was seen in 21% of patients (n = 20) with 18 patients having distant metastatic disease at last follow-up.

Of note, only one study allowed for patients receiving adjuvant RT (the PRADO trial), and all patients with remaining nodal disease had completion LND. In the context of limited follow-up with small patient numbers, patterns of recurrence are just emerging in patients with pNR to neoadjuvant immunotherapy. Huang and colleagues noted 10 patients with recurrences at the last follow-up, with seven having distant metastatic disease [43]. The PRADO trial noted 20 patients with recurrences, with 18 patients having distant metastatic disease at the last follow-up [46]. Although preliminary, these data suggest that anywhere from 10 to 30% of patients will have isolated locoregional recurrences of disease, which may benefit from additional locoregional-based therapy. It is not known if any of the eight patients who received adjuvant RT noted any survival benefits from the RT or if treatment positively or negatively affected their overall quality of life. While neoadjuvant systemic therapy in Stage III melanoma is a promising treatment paradigm, it presents a challenge for how to treat poor responders. The options include the following: continuing prior systemic therapy, shift to second line systemic therapy, post-operative radiation therapy, or surveillance. The role of adjuvant radiation therapy has not been well established; however, there is a phenomenological rationale for thinking that RT can have systemic effects on antitumor immunity by altering the immune microenvironment in ways that may promote an antitumor immune response, and the rate of regional recurrence in these patients is likely high [47,48]. As neoadjuvant therapy continues to be studied/explored further, the optimal clinical management of poor responders will need to be established. This group of patients should be studied to determine a high-risk group for regional failure. Nomograms and multi-factor analysis would be of most benefit to better elucidate these risk factors. Currently, at our institution we give strong consideration to adjuvant radiation therapy in patients with a poor response to neoadjuvant systemic therapy.

## 6. Conclusions

Despite advances in the surgical techniques and available systemic therapy options, regional control of Stage III melanoma remains a clinical challenge. A significant body of evidence strongly suggests that adjuvant radiation therapy reduces the risk of regional recurrence. However, the majority of these data were collected in an era before MAPK pathway inhibitor adjuvant therapy and robust adjuvant immunotherapy options, and no clinical trials have definitively shown a disease-free or overall survival benefit to the use of adjuvant radiation therapy. Future research should refine the definition of high risk for regional recurrence through gene expression profiling or tumor immune profiling scores and elucidate the optimal role of adjuvant radiation therapy in patients treated with neoadjuvant systemic therapy.

## Figures and Tables

**Figure 1 cancers-16-03027-f001:**
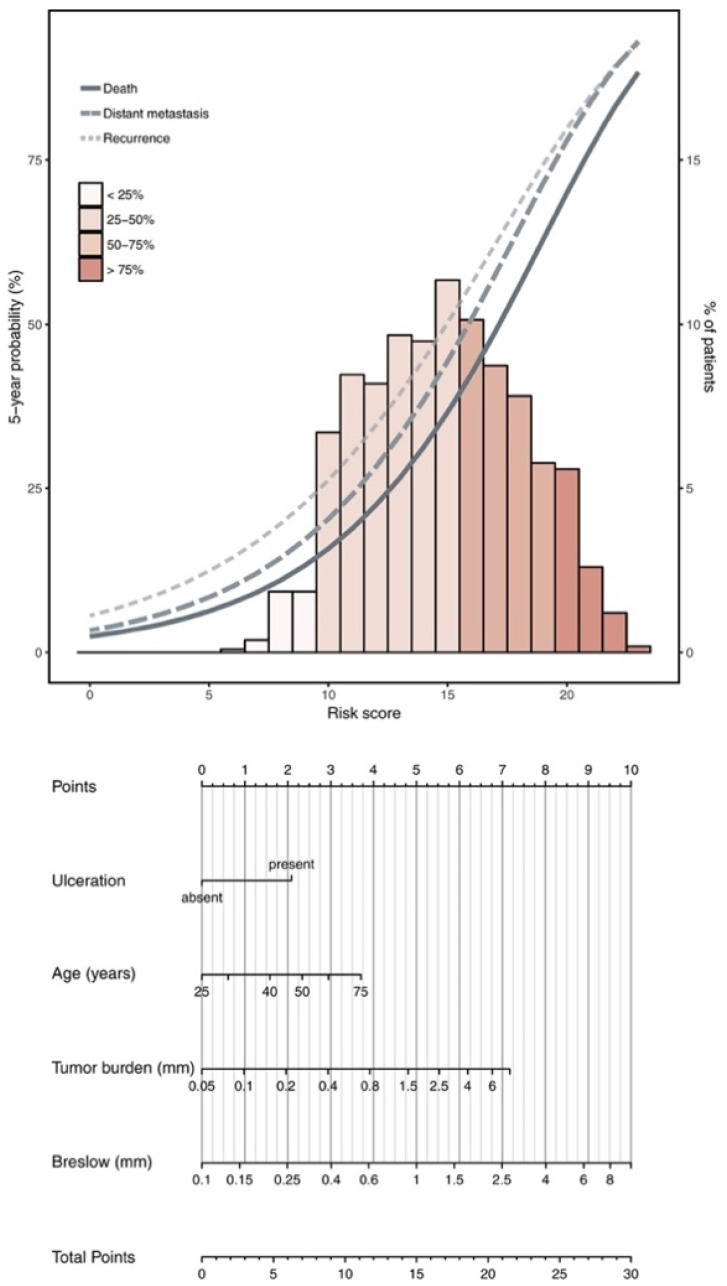
Nomogram calculating risk of recurrence, distant metastasis, and death.

**Figure 2 cancers-16-03027-f002:**
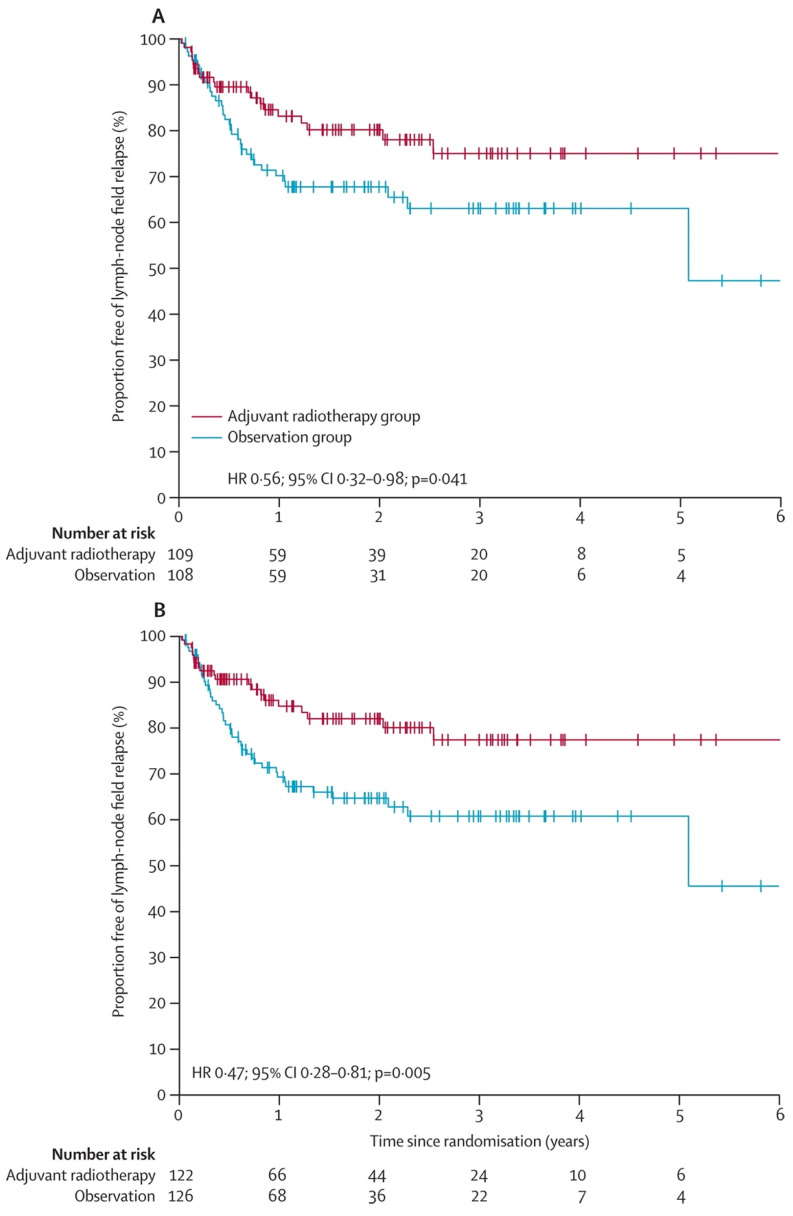
Proportion free of lymph-node field relapse from TROG 02.01 study. (**A**) Time to lymph-node field relapse in eligible population (**B**) Time to lymph-node field relapse in intention-to-treat population.

**Figure 3 cancers-16-03027-f003:**
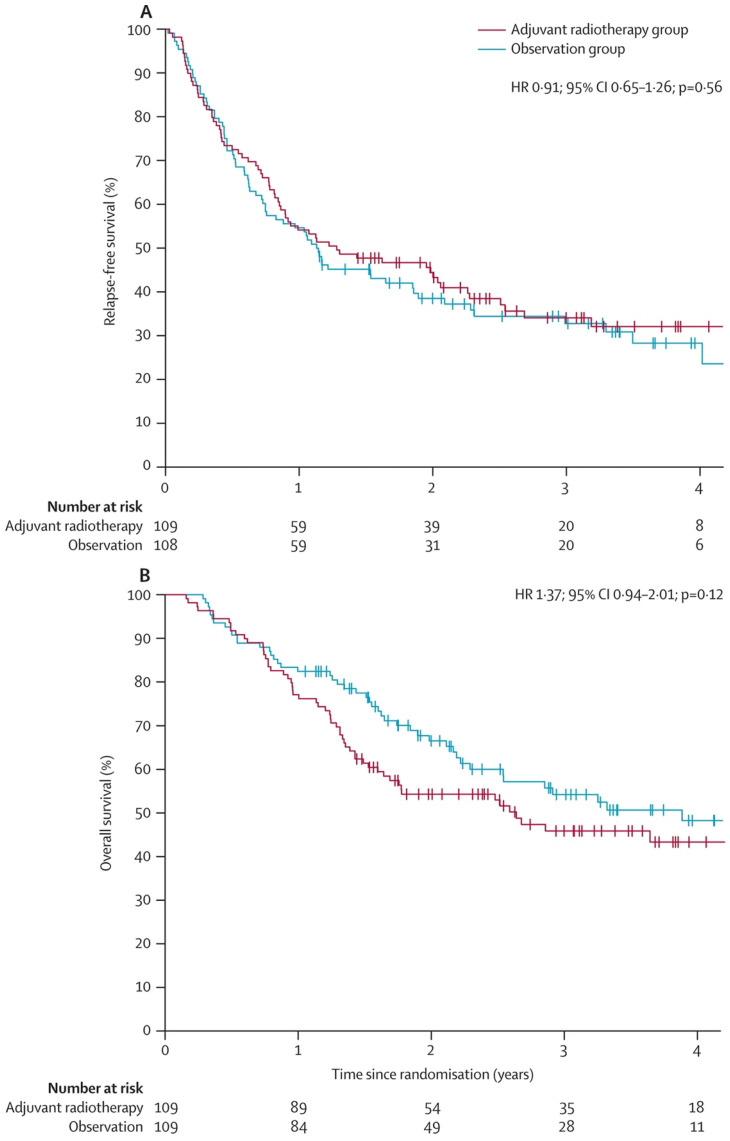
Survival rates from original TROG 02.01 study. (**A**) Relapse free survival (**B**) Overall survival rates.

**Table 1 cancers-16-03027-t001:** Prospective clinical trials investigating RT and ST in melanoma.

NCT Study #	Phase	RT	ST	Primary Endpoint	Patient Population
NCT04594187	2	30 Gy in five fractions	Pembrolizumab, nivolumab	Time to regional nodal recurrence	Node-positive melanoma
NCT05229614	2	Carbon ion RT	Pembrolizumab	Objective response rate (ORR)	Melanoma, non-small cell lung cancer, head/neck squamous cell carcinoma, and urothelial carcinoma
NCT05498805	2	SBRT/hypofractionated	PD-1 inhibitor	Progression free survival	Metastatic melanoma
NCT04318717	2	30 Gy in five fractions	Pembrolizumab	Local tumor control rate	Mucosal melanoma
NCT02392871	1/2	Palliative RT	Combi-AD	Toxicity profile for RT + Combi-AD	Metastatic/unresectable melanoma
NCT04902040	1b/2	Hypofractionated	Plinabulin, pembrolizumab or nivolumab	Incidence of adverse events, ORR	Select advanced cancers
NCT04620603	1/2	Brachytherapy	SOC immunotherapy	Number of participants with tumor response	Stage III and IV melanoma

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
