# Peer review of "Evidence for Radiation Therapy in Stage III Locoregionally Advanced Cutaneous Melanoma in the Post-Immunotherapy Era: A Literature Review"

_cancers, 2024, doi:10.3390/cancers16173027_

Round 1

Reviewer 1 Report

Comments and Suggestions for Authors

This paper, entitled “Treatment of Stage III Locoregionally Advanced Melanoma in the Post-Immunotherapy Era: A Literature Review” is focused on the management of melanoma patients with regional lymph node involvement. 

The topic is of interest, considering the recent possibility of using immunotherapy in the adjuvant setting. However, some critical issues must be resolved before the work can be considered for publication.

Major points:

The prognostic differences based on the different extent of sentinel lymph node involvement (micrometastases, macrometastases, location of metastatic deposits) should be better discussed.

The review is mainly addressed to the role of radiotherapy in the management of stage III melanoma; this should be made clearer in the title and abstract.

In general the work is not easy to read; more subdivision into paragraphs would make it flow more smoothly.

Minor points:

I suggest including “stage III melanoma” as keywords.

Comments on the Quality of English Language

This paper, entitled “Treatment of Stage III Locoregionally Advanced Melanoma in the Post-Immunotherapy Era: A Literature Review” is focused on the management of melanoma patients with regional lymph node involvement. 

The topic is of interest, considering the recent possibility of using immunotherapy in the adjuvant setting. However, some critical issues must be resolved before the work can be considered for publication.

Major points:

The prognostic differences based on the different extent of sentinel lymph node involvement (micrometastases, macrometastases, location of metastatic deposits) should be better discussed.

The review is mainly addressed to the role of radiotherapy in the management of stage III melanoma; this should be made clearer in the title and abstract.

In general the work is not easy to read; more subdivision into paragraphs would make it flow more smoothly.

Minor points:

I suggest including “stage III melanoma” as keywords.

Author Response

Comment #1: The prognostic differences based on the different extent of sentinel lymph node involvement (micrometastases, macrometastases, location of metastatic deposits) should be better discussed.

Response #1: Thank you for pointing this out. I have added a paragraph in the introduction discussing the prognostic differences based on the different characteristics of the sentinel lymph node metastasis. 

Comment #2: The review is mainly addressed to the role of radiotherapy in the management of stage III melanoma; this should be made clearer in the title and abstract.

Response #2: Thank you for comment. I have edited the title to make it more apparent that this is looking at the potential role of radiotherapy in the management of stage III melanoma.

Comment #3: In general the work is not easy to read; more subdivision into paragraphs would make it flow more smoothly.

Response #3: Thank you for your comment. I have divided some of the longer paragraphs into shorter ones to make it easier to read. 

Comment 4: I suggest including “stage III melanoma” as keywords.

Response 4: Thank you for your suggestion. I have added "Stage III melanoma" as keywords. 

Reviewer 2 Report

Comments and Suggestions for Authors

I would recommend to describe much more the body localization of the melanoma and compare with the results.

Other localizations and treatment for melanoma - gyneco loc., epibulbar or uveal loc. should be menshioned in the Discussion

Author Response

Comment 1: I would recommend to describe much more the body localization of the melanoma and compare with the results.

Response 1: Thank you for your comment. The studies discussed in our paper specifically look at patients with cutaneous melanoma. I have updated our abstract and and title to make this more apparent.

Comment 2: Other localizations and treatment for melanoma - gyneco loc., epibulbar or uveal loc. should be menshioned in the Discussion

Response 2: Thank you for your comment. While there are many other forms of melanoma, cutaneous melanoma is the most common form. Other types of melanoma are exceedingly rare and differ from cutaneous melanoma both genetically and in response to treatment. We have updated the title and abstract to make it more apparent that our paper is specifically discussing cutaneous melanoma. 

Round 2

Reviewer 2 Report

Comments and Suggestions for Authors

Pls, give the detail about irradiation techniques, due to the localization e.g. head region etc.

Give detail about mucous, or epibulbar melanoma  and possibilities for treatment

Author Response

Comment #1: Pls, give the detail about irradiation techniques, due to the localization e.g. head region etc.

Response #1: Thank you for your comment. Generally, we do not use radiation to treat primary lesions of cutaneous melanoma (such as on the arm). We are describing the use of radiation therapy as an adjuvant treatment to prevent regional cutaneous melanoma recurrence. Radiation techniques are described in Section 2: Evidence for radiation therapy’s role in regional recurrence risk reduction and differ amongst studies (some use 30Gy in 5 fractions twice weekly and others use 48 Gy in 20 fractions over 4 weeks). The topic of radiation fields and techniques for adjuvant radiation therapy are out of the scope of this paper, as the paper seeks to review existing literature supporting the use of radiation therapy for regional control of cutaneous melanoma. 

Comment 2: Give detail about mucous, or epibulbar melanoma  and possibilities for treatment

Response 2: Thank you for your comment. However, this paper is specifically looking at patients with cutaneous melanoma. Mucosal, epibulbar, and uveal melanoma are all exceptionally rare types of melanoma and are genetically different in both mutations and response to radiotherapy. We have made it clearer in the title and the abstract that this paper specifically looks at patients with cutaneous melanoma.